# Detection of Logos of Moving Vehicles under Complex Lighting Conditions

**Qiang Zhao** [1] **and Wenhao Guo** [2],*

1   School of Environment Science and Spatial Informatics, Suzhou University, Suzhou 234000, China; zq05334@ahszu.edu.cn
2   Faculty of Geosciences and Environmental Engineering, Southwest Jiaotong University, Chengdu 611756, China
*   Correspondence: wh_guo@my.swjtu.edu.cn

**Abstract:** This study proposes a method for vehicle logo detection and recognition to detect missing and inaccurate vehicle marks under complex lighting conditions. For images acquired in complex light conditions, adaptive image enhancement is used to improve the accuracy of car sign detection by more than 2%; for the problems of multi-scale and detection speed of vehicle logo recognition in different images, the paper improves the target detection algorithm to improve the detection accuracy by more than 3%. The adaptive image enhancement algorithm and improved You Only Look One-level Feature (YOLOF) detection algorithm proposed in this study can effectively improve the correct identification rate under complex lighting conditions.

**Keywords:** logo detection; moving vehicles; complex lighting conditions

## 1. Introduction

The detection and classification of image-based vehicle information is an important branch of intelligent transportation systems [1]. As important information on vehicles, their logos are notable and difficult to replace, and correct detection of vehicle logos is helpful for the identification of vehicles [2]. The accurate and rapid identification of vehicle logos has a wide range of applications in vehicle query, vehicle escape, and other aspects; therefore, logo detection and classification are of great significance [3].

When a vehicle is driven on a road, cameras are susceptible to weather conditions when capturing images because the size of vehicle logos in such images is generally not large enough. Generally, traditional research on vehicle logo recognition technology is divided into two parts: positioning and recognition [4]. Vehicle logo detection systems may have a good recognition rate in simple environments with even lighting; however, in the case of a complex environment, dark weather, or uneven lighting, traditional vehicle logo detection systems may hardly reach the standard of practical application. Therefore, in order to recognize vehicle logos on the road, it is necessary to first solve the problem with the influence of complex lighting environments on image detection and then recognize vehicle logos with the help of image enhancement [5].

Image processing methods for complex light conditions mainly include frequency domain processing and space domain processing [6]. Gamma theory [7] and Retinex theory [8] are commonly adopted in spatial domain-based image enhancement methods, and after decades of development, multi-scale image enhancement algorithms [9], etc., have gradually emerged. For automatic recognition of vehicle logos in a complex light environment, adaptive image enhancement algorithms are needed; however, conventional algorithms generally show the disadvantage of halo artifacts at image edges, where the image pixels would change drastically, with wide color distortion. So far, none of the improved methods can satisfy the requirement of adaptive enhancement of complex light [6].

Algorithms based on deep learning have been implemented in the field of logo detection and recognition, with good results achieved [10]. Convolutional Neural Networks (CNNs) [11] can improve the detection rate of vehicle markings by reconstructing the target detection task as a regression issue from image pixels directly to bounding box coordinates and class probabilities. However, in such scenarios as localization, jitters from the motion of the detection devices would result in lower detection accuracy in a moving environment. In addition, when a mature deep learning framework is used to detect logos, there is a problem of missed detection if deep neural network learning is implemented due to different sizes of logos in different images. When the detection target is a moving vehicle, the detection method should also quickly do recognition of the vehicle logo. This is the reason why traditional methods perform automotive logo detection by first locating and then detecting such logos [2].

To address the above issues, this study proposes a method for vehicle logo detection and recognition under complex lighting conditions. Firstly, an adaptive image enhancement algorithm is designed to adjust the brightness and contrast of images so as to reduce the leakage rate in subsequent detection; secondly, a target detection algorithm is improved, and a network structure is designed for the multi-scale problem of vehicle logos in different images; finally, an optimization method is improved to make a balance between the accuracy and speed of vehicle logo detection. Following the above methodological ideas, this study firstly proposes an algorithm for adaptive image enhancement in the principle of gamma transform, and this algorithm can enhance the quality of images under complex light; then, in the process of vehicle logo detection, a darknet algorithm is designed for multi-feature recognition network of different images; finally, an optimization method is proposed corresponding to such features as small display areas of vehicle logos in data as well as high data redundancy.

## 2. Background

The main purpose of logo recognition is to classify logo images. Image classification algorithms generally cover feature representation and classifier-based classification. The initial feature representation usually adopts basic features, such as color, texture, and shape [12], to represent image features.

In the early periods, logo detection in natural scenes was dominated by logo detection in videos [13]. Hollander and Hanjalic [14] proposed a method to determine logo regions through strong contrast between the foreground and background in a color region and realized logo detection by using the template matching in a linear relationship with logos so as to classify logo regions. However, due to strict requirements for logo backgrounds, this algorithm cannot be used for logo detection in complex backgrounds. The traditional local feature points, such as Scale Invariant Feature Transform (SIFT) [15], Histograms of Oriented Gradients (HOG) [16], and Speeded Up Robust Features (SURF) [17], have promoted the rapid development of object recognition, pedestrian detection and face recognition in computer vision processing. For example, Mazzeo proposed a logo detection system—the novel algorithmic pipeline based on SURF combined with RANSAC-lel, which is able to detect multiple logo occurrences in the same image and maintain detection accuracy even under heavy occlusions [18]. Moreover, this algorithm can significantly improve the logo detection effect, but it is very time-consuming. The Retinex algorithm [8], which introduced the wavelet transform, also emerged as a proper balance between image color fidelity and detail enhancement. An improved algorithm based on HSV space [19] can effectively improve the image color distortion; some researchers have improved the algorithm as a whole by introducing the concept of non-neighborhood pixel constraints [19], which converts the image decomposition problem into a quadratic function minimization problem. Nevertheless, this method is time-costly, and its threshold value has to be determined artificially and is not adaptive.

For complex light correction, methods such as gamma transform [19] and histogram equalization [20] are usually utilized. For instance, Arifin [21] proposed a wavelet transform-

based image segmentation method by combining gamma adjustments and transition region filters for image segmentation. Gamma transform is usually used to adjust the overall brightness of an image, but the conventional gamma transform takes the same treatment for all images, so the images of good quality are often overprocessed [22].

Over recent years, with the rapid development of computer vision, deep learning algorithms have become the mainstream in logo detection. CNN algorithms have been successfully applied in general object detection. In the field of logo detection, some scholars have proposed detection methods based on deep learning. For example, Wasin et al. [23] proposed a combination of convolutional neural network and histogram-of-gradients pyramid features for detecting and recognizing vehicle logos from images of the front and rear views of vehicles. Girshick et al. [24] proposed "Regions with CNN" (R-CNN), which has rich feature hierarchies for accurate object detection and semantic segmentation and used selective search (SS) [25] instead of the traditional sliding window method. Mudumbi et al. [26] proposed a method to improve the region proposal network (RPN) by appropriate anchor point selection to do the detection of logos and proposed a modified scheme by combining Faster R-CNN [27] and MobileNet [28], which has an impact on the high-resolution feature maps of mobile devices. Therefore, Oliveira et al. [29] proposed a faster region-based convolutional neural network logo detection algorithm, which selectively extracts candidate frames to achieve multi-scale candidate frame classification. Since then, deep learning algorithms have become popular in the field of computer vision research and scholars have begun to use such algorithms for logo recognition, forming an important branch of computer vision. Bianco et al. [30] put forward a convolutional neural network (CNN) method to replace the traditional feature points; specifically, a convolutional neural network was used to extract the high-level semantic features of logos, and then a support vector machine (SVM) was used to classify such features. Iandola et al. [31] borrowed the GoogleleNet network structure [32] and proposed a different network structure to achieve end-to-end logo recognition, which achieved good recognition results on the public dataset Flickr Logos-32. The detection of moving vehicles' logos is much more complicated than static logo detection. Firstly, as to the quality control of vehicle images captured by cameras in different weathers, the complex light has a great impact on the automatic detection of vehicle logos; and secondly, due to different forms of vehicle logos and different sizes and proportions of vehicles, the detection of vehicle logos becomes challenging.

By improving the quality of data under complex lighting conditions and the You Only Look One-level Feature (YOLOF) [33] algorithm for logo detection, this study will greatly improve the speed and accuracy of logo recognition.

## 3. Methodology

There are three kinds of images in vehicle logo image datasets: low brightness images, high brightness images, and medium (suitable) brightness images. Image enhancement is purposed to reasonably balance the pixel distribution, adjust the brightness and contrast, and improve the distinction between logos and their backgrounds in images under different lighting conditions. Image enhancement can make vehicle logos on outdoor roads more recognizable.

Considering the complexity of vehicle logo sizes and style diversities for logo detection of vehicles in motion, this study improves a target detection algorithm and designs a network structure for the multi-scale problem of vehicle logos in different images.

### 3.1. Adaptive Image Enhancement

This study proposed an adaptive gamma correction method, which can adaptively process logo images at different illumination levels [34]. Firstly, images are converted to HSV color space; then, the V channel is handled by adaptive gamma processing to adjust the image brightness; and finally, the images are converted to RGB color space for

subsequent detection. The image intensity transformation function for image processing used in adaptive gamma correction is as follows (Equation (1)):

$$S = kR^{\gamma} \tag{1}$$

where *s* is the intensity of the output image, *R* is the intensity of the input image, $\gamma$ is the parameter controlling the input-output curve, and *k* is the correction coefficient.

$$I_{cls} = \begin{cases} I_L, 4\sigma \leqslant p \\ I_H, others \end{cases} \tag{2}$$

The standard deviation of the image reflects the dispersion degree of the pixel value and mean value. The larger the standard deviation is, the better the image quality is. Equation (2), $\sigma$ is the standard deviation of an image, and *p* is the contrast coefficient. $I_L$ refers to images with low contrast, while $I_H$ refers to images with medium and high contrast. According to the experimental results, $p = 0.25$ can be used as the criterion for classifying images with different contrast.

The intensity mean of the image reflects the brightness of the image. The larger the mean value is, the higher the brightness of the image is. In this paper, the intensity mean of the image is represented by $\lambda$. Multiple experiments found that when the intensity mean of the image is less than 0.5, the brightness is low, and when it is greater than or equal to 0.5, the brightness is high. Therefore, in this paper, $\lambda = 0.5$ is the threshold to distinguish the brightness of the image. The final image classification category is shown in Table 1.

**Table 1.** Image classification.

| Contrast Category | Intensity Mean ($\lambda$) | Image Category |
|:---:|:---:|:---:|
| $I_L$ | $\geq 0.5$ | Low contrast and high brightness |
| | $< 0.5$ | Low contrast and low brightness |
| $I_H$ | $\geq 0.5$ | High contrast and high brightness |
| | $< 0.5$ | High contrast and low brightness |

For the value of coefficient *k*, different *k* values are used for images with different contrast (Equation (3)):

$$k = \begin{cases} \frac{1}{1+(R_{\gamma}+(1-R_{\gamma})\times \lambda \gamma)}, 0.5 - \lambda > 1 \\ 1 \qquad\qquad, \quad 0.5 - \lambda \leqslant 0 \end{cases} \tag{3}$$

Images in Category $I_L$ have a smaller $\sigma$ value, and most pixels in the image have similar intensity and cluster in a small pixel range. For such images, it is necessary to expand the pixel distribution to a larger range to improve the contrast. In Gamma correction, the larger the $\gamma$ value is, the higher the image intensity and contrast is. Images in Category $I_H$ have a larger $\sigma$ value, and the pixel value is in the dispersed distribution in a dynamic range. Instead of enhancing the contrast, it is more important to adjust the brightness. The slope of the input-output curve is controlled by the $\gamma$ value. The larger the $\gamma$ value is, the higher the image contrast is. For the adaptive Gamma correction proposed in this paper, the following formula is used to calculate the $\gamma$ value for images in Category $I_L$ and $I_H$ (Equation (4)):

$$\gamma = \begin{cases} -\log 2(\sigma), & I_L \\ e^{\frac{1-(\lambda+\sigma)}{2}}, & I_H \end{cases} \tag{4}$$

### 3.2. Improved YoloF Algorithm

Logo detection not only encounters the challenge of complex lighting but also meets the problem with different logo scale sizes. Feature pyramids networks (FPNs) make a great contribution to two-stage object detection. FPNs have two core benefits: first, FPNs

can perform multi-scale feature fusion, i.e., to fuse feature maps at multiple scales to obtain a better representation; second, as a partitioning strategy, FPNs detect targets at different levels of feature maps based on different scales of objects.

The YOLO algorithm is derived from the original YOLOv3 object detection architecture by improving data processing, backbone network, network training, activation function, loss function, and other aspects. It grants the model the optimal performance in matching detection, speed, and accuracy so far [35]. YOLO algorithm firstly extracts the features of an input image through a feature extraction network (also known as the backbone network) and then divides the input image into an *S\*S* grid, and the grid where the object center is located is responsible for object detection. To perform object detection in Category *C*, each grid needs to predict B bounding boxes and conditional probabilities belonging to *C* categories, respectively, and then output the confidence level *Conf(Object)*, which indicates the existence of an object in a bounding box and the accuracy of the output bounding box.

$$IoU = \frac{area(box(Pred) \cap box(Truth))}{area(box(Pred) \cup box(Truth))} \tag{5}$$

$$Conf(object) = Pr(Object) \times IoU \tag{6}$$

*Pr(Object)* signifies whether there is an object falling into the candidate grid. If so, the value of *Pr(Object)* is 1; if not so, the value is 0. *IoU* indicates the intersection ratio between the prediction box and the truth box (Equation (5)), while box (Pred) represents the prediction box and box (Truth) represents the truth box. Each prediction bounding box contains five parameters: *x*, *y*, *w*, *h*, and *Conf(Object)* (Equation (6)). (*x*, *y*) represents the offset between the prediction box center and the truth box center, while (*w*, *h*) represents the width and height of the prediction box.

YOLOF (Figure 1) shows that the most successful aspect of FPN is the strategy of partitioning the optimization problem, rather than multi-scale feature fusion. For optimization, YOLOF introduces an alternative solution without using a complex feature pyramid, but only using a single-level feature map. Based on this simple and efficient solution, a YOLOF detection framework is designed. Experiments on the COCO dataset demonstrate the effectiveness of YOLOF: it can achieve results comparable to the feature pyramid version but 2.5 times faster. In addition, without the Transformer layer, YOLOF is comparable to DETR: YOLOF also uses a single-layer feature map but with seven times fewer training rounds.

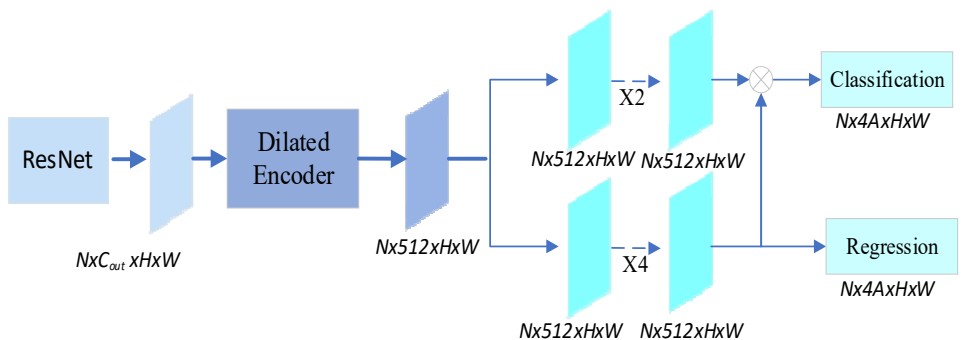

**Figure 1.** The sketch of YOLOF [33].

In Figure 1, 'C5/DC5' represents the output feature of the backbone at a downsample rate of [8,16,32], and '*Cout*' means the number of channels of the feature. This study sets the number of channels to 512 for feature maps in the encoder and decoder. $H \times W$ refers to the height and width of feature maps.

Those who proposed YOLOF believed that the design of loss function is one of the optimization techniques, which can boost the accuracy without increasing the inference time. Traditional object detectors usually use the mean square error to directly regress the center point coordinates, height, and width of the bounding box, while the anchor-based

method estimates the offset. Both of them fail to consider the integrity of the object itself. In the design of the loss function of the original model, the prediction error of the coordinates of the bounding box, the confidence error of the bounding box, and the prediction error of the object classification are taken into consideration.

To solve the multi-scale problem with vehicle logos in moving vehicles' logo images, this study adds a modulation coefficient αto the cross-entropy function based on the idea of Focal Loss [36], which changes the classification error function to Equation (7):

$$L_p = \sum_{i=0}^{S^2} I_{ij}^{obj} \sum_{c=0}^{classes} ([\hat{P}_i^j \log\left(P_i^j\right) + \left(1 + \hat{P}_i^j\right)^\alpha \log\left(1 - P_i^j\right)]) \qquad (7)$$

where $S^2$ represents the number of grids divided by the input image, $I_{ij}^{obj}$ decides whether the *j*-th anchor box of the *i*-th grid is responsible for the judgment of a certain target, *c* represents the category judgment, classes are the category set, $P_i^j$ represents the classification probability, and $\hat{P}_i^j$ represents the estimated classification probability.

Moreover, inspired by Gaussian [5], to improve the accuracy of logo detection, Gaussian distribution is used to improve the loss function and increase the reliability judgment of the bounding box for logo recognition. Taking the x-coordinate of the center point of the bounding box as an example, the modified calculation method of the prediction error of the x-coordinate of the bounding box is as follows (Equation (8)):

$$L_x = -\lambda_{coord} \sum_{i=0}^{S^2} \sum_{J=0}^{B} \log\left(N\left(\hat{x}_i^j \middle| \mu_{t_x}\left(x_i^j\right), \sum_{t_x}(x_i^j)\right) + \epsilon\right) \qquad (8)$$

where, $t_x$ is the offset of the center coordinate of the bounding box from the x-coordinate of the upper left corner of the grid, $\mu_{t_x}$ is the coordinate of the bounding box, and $\Sigma$ is the uncertainty of each coordinate.

In terms of vehicle logo detection speed, the traditional Non-Maximum Suppression (NMS) algorithm arranges the detected target boxes in descending order according to their confidence scores and sets an *IoU* threshold to remove the bounding boxes larger than this threshold until all the predicted boxes are traversed, and the remaining bounding boxes are taken as the final target detection results. Since it is sequential traversal that requires sorting and filtering for each category, it will lead to a loss in the speed of the algorithm. In this paper, we use FAST NMS [37] processing to filter and retain each bounding box, which can strike a balance between vehicle logo detection accuracy and speed.

## 4. Experiment and Analysis

### 4.1. Data

The data in this study come from the Chinese Traffic Sign Detection Benchmark (CCTSDB) produced by Changsha University of Science and Technology [38], available at https://github.com/csust7zhangjm/CCTSDB (accessed on 12 February 2022). The data are expanded on the basis of the Chinese traffic sign dataset (CTSD) and contain a wealth of car information in different contexts, as shown in Figure 2. We selected 3067 images containing 18 common vehicle logos from CCTSDB. Firstly, we filter the images with vehicle logos that meet the complex lighting conditions, then annotate the vehicle logos by labeling, and finally, make the vehicle logo dataset with complex lighting. Data of vehicle logos include: Benz, Volkswagen, Dongfeng, Toyota, BMW, KIA, Peugeot, Hyundai, Haima, Audi, Ford, Citroen, Honda, Wulin, Chevrolet, Chery, Buick, Nissan. The data contains images of vehicle logos in low light, normal light, and strong light conditions.

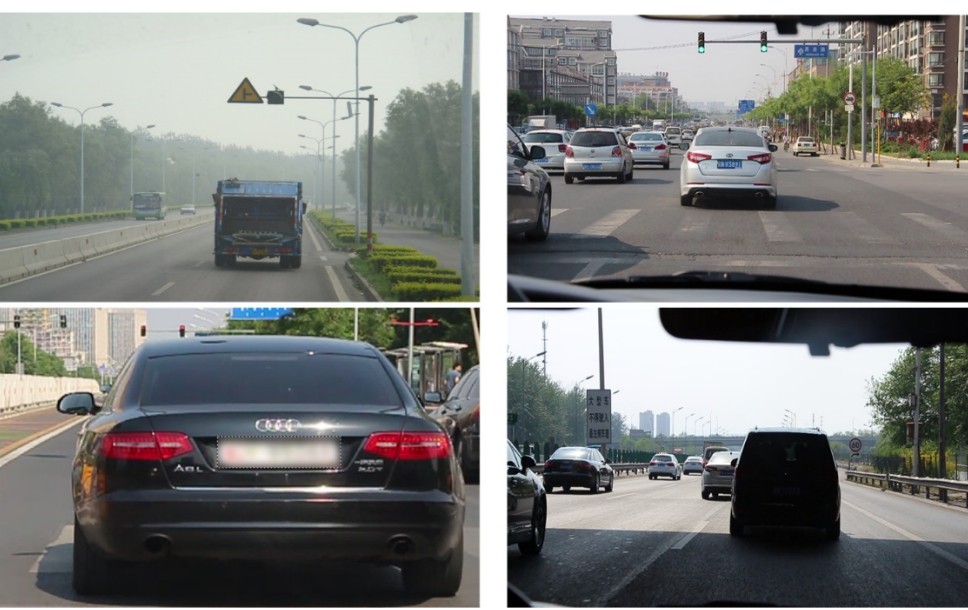

**Figure 2.** Example of CTSD data [38].

*4.2. Results and Analysis*

4.2.1. Image Enhancement

To verify the effectiveness of the adaptive image enhancement algorithm proposed in this study, a comparison test is conducted on certain images containing vehicle logos under different lighting conditions [39].

As shown in Figure 3, we select three representative image enhancement results, images under normal lighting, dark lighting, and strong lighting conditions. As shown in Figure 3a, the comparison reveals that the images in the original CTSD data are not uniformly illuminated but are locally unclear and, therefore, not easily detected and recognized. As shown in Figure 3b, after the adaptive image enhancement, the overall RGB histogram distribution becomes more uniform [40], logos appear prominent, the images' contrast and brightness are significantly improved, the images' details are outstanding, and the original image quality is preserved, thus conducive to logo detection and recognition and thus meeting the requirements of vehicle logo detection in this study [41].

The contrast analysis of images with the above three different lighting conditions demonstrates that the adaptive enhancement algorithm proposed in this study can effectively boost the image quality under complex lighting conditions while providing good samples for subsequent detection, thus conducive to the improvement in detection performance.

4.2.2. Vehicle Logo Detection

The hardware device used in the experiments is installed with a 32G memory, a 2080TI independent graphics card, and a cuda10.2 acceleration framework. Both the original and improved YOLOF networks are used to train the dataset before and after image enhancement with the Ubuntu16.04 operating system and the mmdetection library [42]. The resolution of input images is $128 \times 128 \times 3$. Each experiment is trained for 100 epochs, with a batch size of 64 and a weight decay of 0.0001. Additionally, the learning rate is 0.001. The backbone used for the experiment is resnet 50.

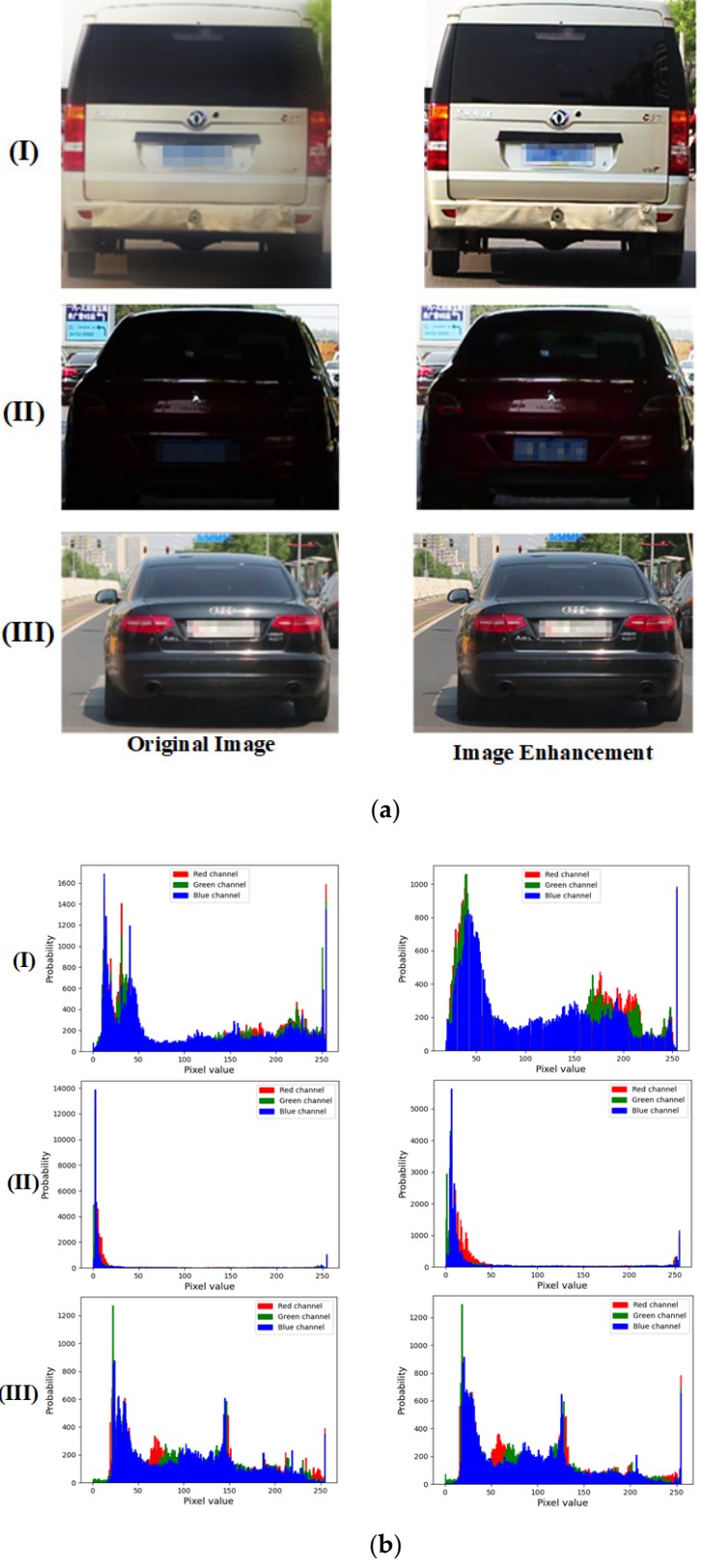

**Figure 3.** Image enhancement in typical scenarios [38]. (**a**) shows the comparison before and after image enhancement in normal lighting conditions, dark lighting conditions and strong lighting conditions, respectively; (**b**) shows the comparison before and after image RGB histogram enhancement in normal lighting conditions, dark lighting conditions, and strong lighting conditions, respectively. (I) is images under normal lighting conditions; (II) is images under dark lighting conditions; (III) is images under strong lighting conditions.

To demonstrate the performance of the improved YOLOF model in vehicle logo detection, this study compares it with the original YOLOF. The experiments use 70% of the image data as the training dataset, 10% as the testing dataset, and 20% as the testing dataset. During the training process, two metrics, i.e., the average precision (AP) and the loss, are obtained from the validation dataset [4].

As shown in Figure 4, the classification accuracy of the improved YOLOF model is boosted by more than 10% compared to the original YOLOF model. As shown in Figure 5, the loss function of the improved YOLOF model is lower and more stable than the original YOLOF model.

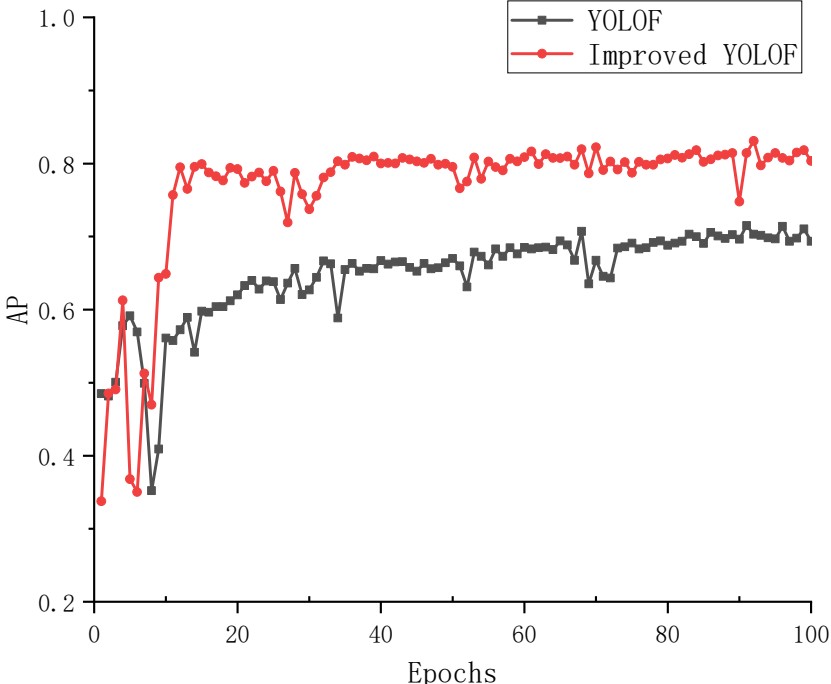

**Figure 4.** Ap of the detection networks.

To verify the accuracy of the improved YOLOF model for vehicle logo image recognition under complex lighting conditions, we compare its detection results with those of the original YOLOF and Faster R-CNN. As shown in Table 2, YOLOF outperforms the Faster R-CNN model for multi-scale vehicle logo image detection [26,43]; compared with the original YOLOF algorithm, the accuracy of the improved YOLOF algorithm is improved by about 3%, therefore effectively improving the accuracy of logo recognition. In addition, the model achieves an increase in detection rate by about 5%.

Overall, by enhancing the loss function model, the improved YOLOF model proposed in this study can boost the detection efficiency and accuracy in vehicle logo detection compared with the original YOLOF model. As shown in Figure 6, the road images acquired under complex lighting conditions are first processed using image enhancement techniques, and then the images are detected using the improved YOLOF network, and finally, the moving vehicle logo results are obtained. The experimental results show that our proposed method can effectively accomplish the recognition of car markers under complex lighting conditions.

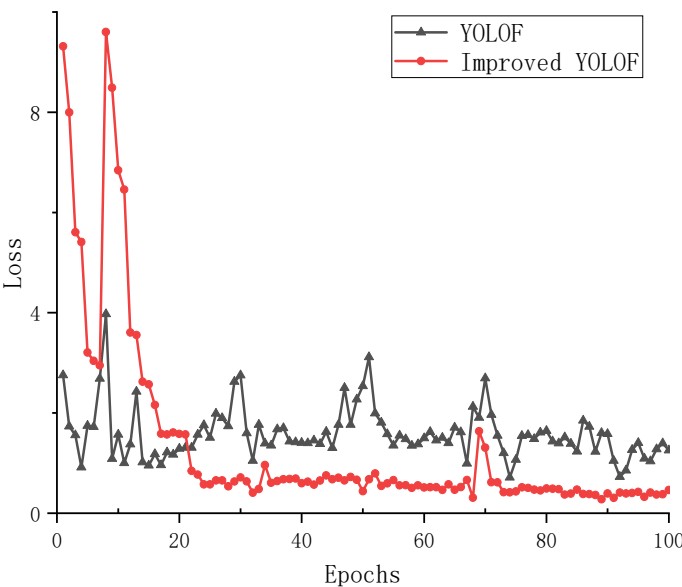

**Figure 5.** Loss of the detection networks.

**Table 2.** Comparison of detection results between the original YOLOF and improved YOLOF.

| Data | Algorithm | Number of Pictures | FPS | AP(%) |
|---|---|---|---|---|
| Original data | Faster R-CNN | 3067 | 38.3 | 80.2 |
| | Original YOLOF | | 40.5 | 89.27 |
| | Improved YOLOF | | 45.05 | 92.43 |
| Corrected data | Faster R-CNN | 3067 | 38.4 | 83.5 |
| | Original YOLOF | | 40.5 | 93.7 |
| | Improved YOLOF | | 45.07 | 95.86 |

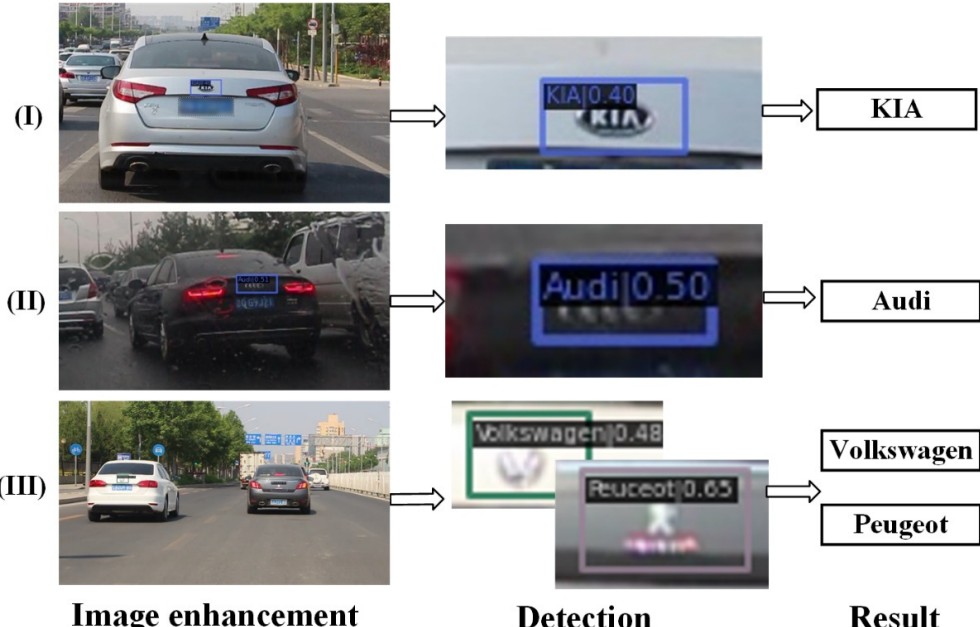

**Image enhancement**  **Detection**  **Result**

**Figure 6.** Results of moving vehicles' logo detection [38]. (I) is images under normal lighting conditions; (II) is images under dark lighting conditions; (III) is images under strong lighting conditions.

## 5. Conclusions

In view of the fact that commonly used object detection frameworks fail to solve the problem of vehicle logo detection and recognition under complex lighting conditions, this study puts forwards a logo detection and recognition method based on image enhancement technology and an improved YOLOF framework. The experimental results from the comparison between the proposed method and the original YOLOF algorithm demonstrate that: (1) The adaptive image enhancement algorithm proposed in this study can make reasonable adjustments to image samples under complex lighting conditions while reducing the rate of missed detection of subsequent detection algorithms; (2) The detection accuracy and speed of the trained detection model are improved by boosting the error loss function of YOLOF and the NMS algorithm; (3) The comparative experimental analysis of the detection performance of the proposed method and the original YOLOF algorithm proves that better detection results can be delivered by decomposing the logo detection under complex lighting conditions into two tasks: image enhancement and logo detection. This approach provides another solution to vehicle logo detection and recognition. In future work, the ablation study will be used as a research direction for vehicle logo detection and unravel the effects of image enhancement, focal loss, Gaussian loss, and NMS on instances of different sizes.

**Author Contributions:** Conceptualization, Q.Z. and W.G.; methodology, Q.Z.; software, Q.Z.; validation, Q.Z.; formal analysis, Q.Z.; investigation, Q.Z.; writing—original draft preparation, Q.Z.; writing—review and editing, W.G.; visualization, Q.Z.; supervision, Q.Z. All authors have read and agreed to the published version of the manuscript.

**Funding:** This research received no external funding.

**Institutional Review Board Statement:** Not applicable.

**Informed Consent Statement:** Not applicable.

**Data Availability Statement:** The data presented in this study are available in the submitted article.

**Conflicts of Interest:** The authors declare no conflict of interest.

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
