# Peer review of "Detection of Logos of Moving Vehicles under Complex Lighting Conditions"

_applsci, doi:10.3390/app12083835_

Round 1
Reviewer 1 Report
Dear Author,
The manuscript "Detection of Logos of Moving Vehicles Under Complex Lighting Conditions" by Qiang Zhao and Wenhao Guo discusses the YOLOF and improved YOLOF model application in detection process of logo of moving vehicles. The structure of the article is clear and logical. The Author determined the need for development the algorithm for car logos detection in different lighting conditions. However, in my opinion the manuscript needs to be improved in some fields.
The Abstract section should present quantitative results and not only the most important qualitative results and/or generic considerations. The specific research results contained in the article are missing. Moreover, in my opinion, the structure of the abstract should be changed, in particular the fragment "Firstly, an adaptive image enhancement algorithm is designed to adjust the brightness and contrast of images, so as to reduce the leakage rate in subsequent detection; secondly, a target detection algorithm is improved and a network structure is designed for the multi-scale problem with vehicle logos in different images; finally, an optimization method is improved to strike a balance between vehicle license plates' recogniznition accuracy and speed. " should be divided into separate opinions. Therefore, significant improvements are expected in this part of the manuscript.
In line 42, the Authors use the personal form ("…however, they..."). This is not correct in high-quality articles. It suggests modifying this part of the article. Please check the entire article in terms of personal form. The same remark is for line 131, 282 nad 286.
The test results available in the literature should be specifically referred to Authors results. When the results are not discussed and conveniently supported by the open literature, questionable conclusions are obtained. Currently, the article looks more like a report from test than a scientific article. Please modify 4.2.1 paragraph.
Research articles should present the directions of further research. I suggest adding one paragraph in the conclusion chapter.
Specific remarks/editorial comments/typos:
- line 113 – there is “…Zhan [26]proposed…”, should be [26] proposed,
- line 140 – there is “…Feature(YOLOF)…”, should be Feature (YOLOF),
- line 235 – there is “…{8,16,32}, and…”, should be [8,16,32],
- line 266 – there is “…Technology[37],…”, should be Technology [37],
- Figure 3 – the charts in Figure 3 are invisible, please enlarge or delete them entirely. If the figure contains sub-drawings a), b) c), they should be described in the drawing caption. Please complete this.
- Line 320 – Table 1 was in line 180. Please change the number of table in line 320 to number 2,
- Figure 6 – the same remark as to Figure 3, there are some sub-figures, please add detailed caption,
Major corrections should be implemented before considering the work for publication. I hope these suggestions can help to improve the quality of this paper.
I wish you all the best.
Author Response
Dear Reviewer,
Thanks for your comments concerning our manuscript entitled “Detection of Logos of Moving Vehicles Under Complex Lighting Conditions” (Manuscript ID: applsci-1644159). Those comments are all valuable and very helpful for revising and improving our paper, as well as the important guiding significance to our research. The modifications mentioned are marked in RED in the revised manuscript.
Sincerely yours,
Wenhao Guo
E-mail: wh_guo@my.swjtu.edu.cn

Reviewer 2 Report
Summary of the paper
- Authors present a car logo detection application using a two step approach: image preprocessing and improved You Only look One-level Feature (YOLOF) detector.
- The image preprocessing is a gamma correction in a "power law" form with four presets of (k, gamma) corresponding to four types of images that can happen: low/high average brightness and low/high contrast.
- The improvement in YOLOF consists of adding a Focal Loss (RetinaNet paper) and Gaussian parameterization of bounding box predictions (Gaussian YOLO paper).
- Authors then demonstrate superior performance in both speed and average precision on a custom dataset (see below) of images mostly captured by dashboard cameras on cities' streets.
Comments
- When I started reading the paper, it was not clear to me that the target application actually is logo detection. For example, in abstract, you mention "optimization method is improved to strike a balance between vehicle license plates’ recognition accuracy and speed", even though the paper does not address this problem at all. Similarly, license plate recognition is mentioned several times in the Introduction and even the "Methodology". Why? Did you switch the problem during writing of the paper?
- The part of the "Background" section that covers detectors based on deep learning is not structured particularly well. Reading about AlexNet while the reference cites ResNets and all that after already introducing Fast R-CNN? Also, the text constantly jumps back and forth from generic object detection to logo detection.
- The section "Methodology" is quite clear and understandable, but I suspect also incomplete. There are supposed be two key proposals to the paper: image enhancement and YOLOF improvement. There are two improvements in YOLOF explained in the Methodology section: adding focal loss and gaussian parameterization to the loss function, none of which having any effect on inference computational complexity. However, in the section "Experiment and Analysis", the resulting Improved YOLOF is actually faster than the original YOLOF. Moreover, you mention "boosting the loss function of YOLOF and the NMS algorithm" in the "Conclusion" section. I think the "Methodology" section should explain the "boosting of NMS" as well.
- The data used in the "Experiment And Analysis" is, unfortunately, only partially available. Only images with traffic sign annotation can be downloaded. How did you get the vehicle logo annotations? Are the data available?
- It would be nice too see an ablation study in the "Experiment And analysis", i.e. decouple how each of the improvements (image enhancement, focal loss, gaussian loss, NMS) influences the average precision (AP). Also, why do you think the focal loss helped in your case when it failed in YOLOv3?
- I would also suggest more metrics to monitor, such as AP for small, medium and large instances.
- How many parameters does your model have? What is the backbone? Have you tried different backbones to balance the model complexity and accuracy? Is the code for the paper available somewhere? Can anyone reproduce the results?
Author Response

(The authors gave the same response as above.)

Round 2
Reviewer 1 Report
Dear Authors,
thanks for addressing the comments. However, in my opinion, the manuscript needs further improvement as not all issues have been modified.
The Authors changed the abstract as I suggested. Nevertheless, in the new version, the personal form "we use adaptive" was once again used. This part of the manuscript needs to be changed. The same remark is for the 452 line and new part of the manuscript in which it is described future work direction.
Authors' response to Concern # 3 :, contained in the file applsci-1644159-coverletter is insufficient. I understand the description that came with the manuscript. However, it describes only the obtained results. These results should be compared with the specific results of other authors with reference to specific publications in references. This remark was not taken into account by the Authors. In chapter 4.2. Results And Analysis does not make any reference to the results of research by other authors included in references. Please add 3-5 or more References with previous work which will prove that the performed research and the obtained results bring new knowledge in the subject of the article.
I wish you all the best.
Author Response
Dear Reviewer,
Thanks for your comments concerning our manuscript entitled “Detection of Logos of Moving Vehicles Under Complex Lighting Conditions” (Manuscript ID: applsci-1644159). Those comments are all valuable and very helpful for revising and improving our paper, as well as the important guiding significance to our research. The modifications mentioned are marked in RED in the revised manuscript. The following is a point-to-point response to the comments of editors and reviewers:
Sincerely yours,
Wenhao Guo
E-mail: wh_guo@my.swjtu.edu.cn

Reviewer 2 Report
- The paper still mentions license plate recognition on several occasions, e.g. lines 29, 78, 335. It's fine as a related area, but "making/striking a balance ..." (a formulation repeated multiple times throughout the paper) as if it were a target application and one of the goals of the paper?
- The structure of "Introduction" has been improved, although still doesn't read particularly well. For example, text on line 123 mentions logo detection based on R-CNNs before citing their proposal by Girshick. By the way, item [3] of the bibliography cites "Open logo detection challenge" by Su et al., so it does not correspond to the text referencing it (Bao et al., who is not in the "References" at all).
- Your response to #4, which perfectly and clearly explains everything about the data preparation, should simply be part of the paper.
- The ablation study comment has been somewhat disappointingly dealt with by suggesting it as a future work in the "Conclusion" section. While I don't believe it's strictly necessary, as the experiments decouple at least the data correction and the model improvement, high quality research should be as detailed as possible to better understand the effects of individual components and to see what works and what doesn't.
- Cite the mmdetection library or at least mention it as a footnote. Not only is it a common courtesy, but also it improves reproducibility, especially when your codes are not released along with the paper.
- There is a typo on line 453: "focus loss".
Author Response

(The authors gave the same response as above.)

Round 3
Reviewer 1 Report
Dear Authors,
thanks for addressing the comments. The article "Detection of Logos of Moving Vehicles Under Complex Lighting Conditions" can be publish in Applied Sciences journal.
Best regards